# Taxonomy, Phylogeny, and Size Evolution in the Spider Genus *Megaraneus* Lawrence, 1968 (Araneae: Araneidae)

**DOI:** 10.3390/insects16100992

**Published:** 2025-09-24

**Authors:** Klemen Čandek, Eva Turk, Pedro de Souza Castanheira, Kuang-Ping Yu, Matjaž Gregorič, Volker W. Framenau, Ingi Agnarsson, Matjaž Kuntner

**Affiliations:** 1Department of Organisms and Ecosystems Research, National Institute of Biology, 1000 Ljubljana, Slovenia; klemen.candek@nib.si (K.Č.); eva.turk@nib.si (E.T.); kuang-ping.yu@nib.si (K.-P.Y.); 2Harry Butler Institute, Murdoch University, Murdoch, WA 6150, Australia; pedro.castanheira@murdoch.edu.au (P.d.S.C.); volker.framenau@murdoch.edu.au (V.W.F.); 3Biotechnical Faculty, University of Ljubljana, 1000 Ljubljana, Slovenia; 4Jovan Hadži Institute of Biology, Research Centre of the Slovenian Academy of Sciences and Arts, 1000 Ljubljana, Slovenia; matjaz.gregoric@zrc-sazu.si; 5Postgraduate School ZRC SAZU, 1000 Ljubljana, Slovenia; 6Department of Terrestrial Zoology, Western Australian Museum, Welshpool, WA 6986, Australia; 7Zoological Museum Hamburg, Leibniz Institute for the Analysis of Biodiversity Change (LIB), Centre for Taxonomy & Morphology, 20146 Hamburg, Germany; 8Faculty of Life and Environmental Sciences, University of Iceland, 102 Reykjavik, Iceland; iagnarsson@hi.is; 9Department of Entomology, National Museum of Natural History, Smithsonian Institution, Washington, DC 20024, USA; 10Centre for Behavioural Ecology and Evolution, College of Life Sciences, Hubei University, Wuhan 430061, China

**Keywords:** sexual size dimorphism, backobourkiines, trait evolution, South Africa

## Abstract

Many spider species show large differences in size between males and females, but biologists still do not fully understand how or why this evolves. In this study, we focused on the systematics and evolution of a little-known African spider, *Megaraneus gabonensis*, in which females are about four times larger than males. We provided an overdue taxonomic description of the species and examined where the genus *Megaraneus* fits in the spider tree of life. Our findings show that this spider is a part of a clade that has previously been thought to live only in East Asia and Australia, but we now know it also occurs in Africa. We also found that in *Megaraneus*, the large size difference between the sexes resulted primarily because females became larger, not because males would become smaller. This research helps us better understand how spider body size evolves differently in males and females and highlights the importance of studying lesser-known species to fill in knowledge gaps about the natural world.

## 1. Introduction

Spiders exhibit some of the most striking examples of female-biased sexual size dimorphism (SSD) in the animal kingdom [1,2,3,4,5,6]. In some orb-weaving species, females are more than twice the linear size of males, a pattern termed extreme sexual size dimorphism (eSSD) [1]. Phylogenetic studies have inferred between four [4] and nine [7] independent evolutionary origins of eSSD in araneoid spiders. Early phylogenetic research on male and female evolutionary size changes focused on whether eSSD in particular clades is better attributable to patterns of “male dwarfism” versus “female gigantism” [4,8]. Recent studies instead suggest that sex-specific size evolution is often more complex and lacks a universal pattern across clades [9].

Despite recent advances, the selective forces driving SSD in spiders remain incompletely understood. Several hypotheses have been proposed, emphasizing different interactions of natural and sexual selection, ecological pressures, and sexual conflict (reviewed in [1]). Female gigantism is associated with fecundity selection whereby larger body size allows for greater egg production and increased reproductive success [10,11,12]. In contrast, small male size may result from selection for increased agility, more efficient mate searching, and improved predator avoidance [2,13,14,15]. To explain the evolution of SSD in spiders, Kuntner & Coddington [1] built on previous work [2,3,5] to propose a differential equilibrium model [16], where reinforcing and opposing selection pressures act together and where the direction and magnitude of SSD vary across lineages.

A major obstacle to understanding SSD evolution across spiders is incomplete taxonomic sampling and poor understanding of intra- and interspecific size variation. Many species are known only from one sex and are described based on minimal morphological data, often without intraspecific variation or ecological context [17]. For example, some of the best-known examples of eSSD in spiders are found in the family Araneidae [1], whose significant portion of species diversity remains underexplored, particularly in the tropics. Continued field research in poorly studied regions is essential for discovering new species, documenting SSD, and refining phylogenetic frameworks that underpin evolutionary inference.

In South Africa, we documented the natural history of *Megaraneus gabonensis* (Lucas, 1858), a conspicuous (Figure 1) but poorly known araneid. Its eSSD is particularly striking, with males only about a fourth the body length of females (Figure 1A). This eSSD case has remained elusive as over 100 years passed since the description of the species to the discovery of its male. When Lawrence [18] described the male, he noted the dimorphism in “[t]he genus, with regard to the discrepancy of form and size in the sexes, agrees more with *Argyope*, *Cyrtophora*, *Caerostris* and *Gasteracantha* than with *Araneus*…” (p. 110) and “[t]here is a much greater discrepancy of size in the male and female than in the case of *Caerostris*…” (p. 114). Further males have never been reported, and the genus and species lack recent taxonomic treatment. In part due to the convoluted taxonomic history of *M. gabonensis*—originally described as *Epeira*, and transferred to *Cyrtophora* Simon, 1864 and *Caerostris* Thorell, 1868—the genus has never been placed phylogenetically.

We here provide these missing links for a more comprehensive understanding of araneid SSD evolution by redescribing *M. gabonensis* based on both sexes, reconstructing its phylogenetic relations to other araneids, and investigating the evolutionary origins of its eSSD. Specifically, we ask: (i) Where does *Megaraneus* fit within the broader araneid phylogeny? (ii) Does eSSD in *Megaraneus* represent an independent origin, or is it part of a shared evolutionary trajectory with other eSSD araneid lineages? (iii) Can we attribute the observed eSSD in *Megaraneus* to female gigantism, male dwarfism, or a more complex interplay of sex-specific selection pressures?

## 2. Materials and Methods

### 2.1. Fieldwork

Two adult male and twelve adult female specimens of *Megaraneus gabonensis* were collected in 2019 at multiple sites within iSimangaliso Wetland Park, KwaZulu-Natal Province, South Africa (see Appendix B for details). Specimens were hand-collected during diurnal and nocturnal surveys and preserved in 70% ethanol for morphological examination. One leg of each specimen was removed and preserved in 96% ethanol for molecular analyses.

### 2.2. Microscopy and Imaging

Micrographs were taken at the Harry Butler Institute, Murdoch University (Australia), using a Leica DMC4500 digital camera (Leica, Wetzlar, Germany) mounted on a Leica M205C stereomicroscope. Images were captured in multiple focal planes and focus-stacked using Leica Application Suite X v.3.6.0.20104. Final image editing and figure assembly were carried out in Adobe Photoshop CC 2023. Color patterns of described specimens were based on specimens preserved in 96% ethanol. The protocol to expand the male pedipalp and to clarify the female genitalia was carried out by alternately submerging these structures for around 5 to 10 min in 10% KOH and distilled water until the pedipalp was fully expanded and the female genitalia sufficiently clarified with internal ducts and spermathecae visible.

### 2.3. Molecular Procedures

We extracted DNA from all 14 specimens using the DNeasy Blood & Tissue Kit (Qiagen, Hilden, Germany), following the manufacturer’s instructions. PCR protocol, including primers used, followed the procedure outlined in Yu et al. [19]. From one male and one female, we targeted five standard molecular markers: mitochondrial cytochrome c oxidase subunit I (*COI*) and *16S* rRNA, as well as the nuclear ribosomal *18S*, *28S* rRNA, and Histone *H3*. From all remaining specimens, we only amplified *COI*. PCR products were purified and sequenced by Macrogen Europe B. V. (Amsterdam, The Netherlands). Sequences were assembled and edited using Geneious Pro v.5.6.7. and deposited in GenBank (for accession numbers see Appendix B).

### 2.4. Phylogenetic Reconstruction

To infer the phylogenetic placement of *Megaraneus*, we aligned the new sequences to the dataset of Scharff et al. [20], which includes 133 araneid taxa. To further increase taxon sampling we added newly available sequences of *Cyphalonotus* [19], belonging to another lineage containing a case of eSSD. The concatenated matrix is available as Appendix A. Alignment was performed in MAFFT v.7 [21] using default settings and concatenated in Mesquite v.3.8.1 [22]. We reconstructed phylogenetic relationships using MrBayes v.3.2.7 [23], run via the CIPRES Science Gateway [24]. We used the generalized time-reversible model with gamma distribution and invariant sites (GTR + G + I) for each partition, following Scharff et al. [20]. We ran two independent runs, each with four MCMC chains, for 30 million generations, with a sampling frequency of 1000 and relative burn-in set to 25%. The starting tree was random [20].

To further corroborate the phylogenetic placement of *Megaraneus* as suggested by the result of the Bayesian analysis, we ran a second, Maximum Likelihood phylogenetic analysis on a representative subset of 63 taxa. We used IQ-TREE 2 v.3.0.1 [25] with a partitioning scheme from Scharff et al. [20]. Substitution model for each partition was determined by IQ-TREE 2. Bootstrap support was assessed with 1000 ultrafast bootstrap replicates [26,27] and SH-like approximate likelihood ratio test [28].

In order to evaluate evolutionary trends of sex-specific size evolution trajectories within the focal *M. gabonensis* group in downstream analyses, we reconstructed a separate phylogeny only including taxa with available body size measurements (see Section 2.6
*Trait evolution*). We expanded the focal group by adding *Backobourkia brounii* (GenBank accession sode: FJ873121) and *Backobourkia collina* (GenBank accession code: FJ873123), and eliminated representatives of *Acroaspis* and *Carepalxis*, for which such data is unavailable. Phylogenetic reconstruction used the same settings as above.

### 2.5. Morphological Examination

Measurements were taken under a Keyence VHX-7000 digital microscope and are reported in millimeters. Body and appendage measurements were recorded to the nearest 0.1 mm, while finer structures such as the eyes and labium were measured to the nearest 0.01 mm for greater precision. Terminology follows standard conventions in recent ‘backobourkiines’ systematics [29,30,31,32,33]. The terminology of the views of the male pedipalp considers its position as a limb, i.e., the full view of the bulb with the cymbium in the background is retrolateral (not ventral) because in Araneidae, the pedipalp is twisted so that the cymbium is situated mesally [32,33]. The epigyne has two main parts, the base (encapsulating the internal genitalia) and the scape. We refer to the central part of the base in ventral view as atrium, which becomes the central division in posterior view [29,30,31].

### 2.6. Trait Evolution

For comparative analyses, we compiled body length data for taxa, reconstructed as members of the focal *Megaraneus* clade, from the literature (Appendix A). SSD was calculated as the ratio of female to male body length, following Kuntner & Coddington [1]. We projected continuous trait data (body length and SSD) onto our focal clade phylogeny in Mesquite v.3.8.1 [22] using squared-change parsimony, requiring no assumptions about the underlying statistical distribution of trait evolution. This allowed a descriptive assessment of general trends in trait evolution within the clade, including ancestral state estimation and size trajectory comparisons for: (1) male body length vs. SSD, (2) female body length vs. SSD, and (3) male body length vs. female body length.

## 3. Results

The freshly collected material allows for thorough taxonomic treatment of *M. gabonensis* and, by current monotypy, of the genus *Megaraneus* (see Section 5, Taxonomy). Analyses of body length data suggest that *M. gabonensis* females are between 3.4 and 5.1 times larger than males, with an average SSD value of 4.1.

### 3.1. Phylogenetic Placement

The reconstructed Bayesian araneid phylogeny generally exhibits low support for deeper nodes, and the recovered topology deviates from that of Scharff et al. [20]. Nevertheless, it places *M. gabonensis* within a well-supported clade informally named ‘backobourkiines’ *sensu* Scharff et al. [20] (Figure 2A; Appendix A). Within ‘backobourkiines’ (Figure 2B; Appendix A), *Megaraneus* is sister to a clade containing *Backobourkia* and *Parawixia dehaani* (but see Discussion for a comment on probable generic misattribution), with the latter and *Backobourkia collina* exhibiting eSSD. The remaining two *Backobourkia* species exhibit modest SSD, reflecting intrageneric variation. Other genera within the broader ‘backobourkiines’ assemblage, i.e., *Novakiella*, *Socca*, *Salsa*, *Plebs*, *Hortophora*, and *Singa* (but see Results section *Size evolution*), are all sexually size monomorphic.

The Maximum Likelihood phylogeny corroborates the placement of *Megaraneus* within ‘backobourkiines’ (Appendix A) and as sister to *Parawixia dehaani* + *Backobourkia* with high support (98/99).

### 3.2. Size Evolution

Ancestral state reconstruction of body length and SSD across the ‘backobourkiines’ clade rejects an independent origin of eSSD in *M. gabonensis*. The common ancestor of *Megaraneus*, *Backobourkia*, *Parawixia dehaani*, *Hortophora*, *Plebs*, *Salsa*, *Socca*, *Singa*, and *Novakiella* is reconstructed to already exhibit SSD values above the 2.0 threshold, typically used to define eSSD (Figure 3). Although not directly relevant for the present study, note that the placement of *Singa nitidula* within the ‘backobourkiines’ in Scharff et al. [20] and other recent, yet unpublished phylogenies [34], is inconsistent and thus debatable.

Within this clade, SSD trajectories show divergent trends. Except for *Plebs bradleyi*, the lineage containing *Hortophora*, *Plebs*, *Salsa*, *Socca*, *Singa*, and *Novakiella* has seen a decline to SSD values below 2.0. In contrast, the sister clade comprising *Megaraneus*, *Backobourkia*, and *Parawixia dehaani* exhibits increases in SSD values beyond the ancestral state. The highest reconstructed values occur in *Parawixia dehaani* (SSD = 4.4), followed by *M. gabonensis* and *Backobourkia collina* (both SSD = 4.1), while the other two species of *Backobourkia* show a decrease in SSD (see Appendix A).

Separate reconstructions of male and female body length (Figure 3) indicate that female body length has increased towards *Megaraneus*, whereas male body length has remained relatively unchanged from the ancestral values. This directional asymmetry in trait evolution underlies the observed increases in SSD toward the terminals.

## 4. Discussion

By integrating morphological variation, taxonomy, phylogenetics, and trait reconstruction, we shed new light on the evolutionary history of *Megaraneus gabonensis*, the sole representative of its genus, and its placement within Araneidae. Prior descriptions of the genus and species are, by today’s standards, outdated. We therefore provide a redescription (see Section 5, Taxonomy) that adheres to modern standards in araneology [17]. The major taxonomic contribution of our present description is the inclusion of microscopic imaging that shows detailed morphological features used in genus and species diagnoses.

Our original phylogenetic placement of *Megaraneus* recovers it nested within ‘backobourkiines’ *sensu* Scharff et al. [20], a clade previously thought geographically restricted to East Asia and Australia. This unexpected phylogenetic affinity significantly expands the known biogeographic range of the clade to include the Afrotropics and raises the question of whether the current distribution reflects an isolated case of long-distance dispersal or undersampled diversity in Africa.

Although the ‘backobourkiines’ portion of the tree is well supported, many deeper nodes of the araneid tree of life are not. Scharff et al. [20] reconstructed the araneid phylogeny using five Sanger markers, unsurprisingly not fully resolving relationships within an ancient family of over 100 million years [35,36]. The limits of Sanger data and the lack of sufficient detail on phylogenetic methodology used by Scharff et al. [20] are mirrored in our inability to recover their deeper topology. These topological uncertainties limit broader inferences about the evolutionary origins of SSD across the family. We therefore limit our discussion to the part of the topology relevant for *Megaraneus* and the size evolution within its immediate phylogenetic proximity.

SSD within the ‘backobourkiines’ is variable, ranging from minimal to eSSD. Our ancestral trait reconstructions indicate that eSSD was likely present in the common ancestor of *Megaraneus*, *Backobourkia*, and *Parawixia dehaani*. It should be noted, however, that the latter species is a probable misattribution to the genus *Parawixia* and in fact constitutes a separate genus [20].

These reconstructions suggest a broad pattern of fluctuating SSD across the clade, with independent trajectories of male and female size change and thus SSD magnitude. These trajectories include secondary reductions in *Hortophora* and *Novakiella* and further increases in *Megaraneus* and *Parawixia*. In *Megaraneus*, the observed pattern of sex-specific size changes suggests that its eSSD is a case of female gigantism [8], given that male body length has remained conserved, but the female body length has steadily increased. Note, however, that the observed trait evolution patterns are based only on ‘backobourkiines’ with available size data, omitting genera such as *Acroaspis* and *Carepalxis*. Additional sampling is thus necessary to further test our interpretations.

In conclusion, the pronounced eSSD in *Megaraneus* (Figure 1A) is not an isolated phenomenon, but rather part of a more complex and dynamic pattern of size evolution within a phylogenetically coherent group. Within *Backobourkia* alone, SSD varies from extreme to only slight; moreover, the eSSD species *B. collina* is significantly smaller overall and occupies a different habitat (grasslands vs. trees) compared to its congeners, making it an interesting object of future exploration. Results across the clade are consistent with the view that both female gigantism and male dwarfism have contributed to spider SSD across evolutionary timescales [1,37], and that the intensity and direction of size divergence are lineage-, or even species-specific, possibly shaped by ecological and reproductive factors [9].

## 5. Taxonomy


**Order Araneae Clerck, 1757**

**Family Araneidae Clerck, 1757**

**Genus *Megaraneus* Lawrence, 1968**

** **
** **
**Type species.** *Megaraneus gabonensis* (Lucas, 1858); by monotypy.
** **
** **
**Diagnosis.** Somatically, specimens of *Megaraneus* resemble those of the phylogenetically close *Backobourkia* Framenau, Dupérré, Blackledge & Vink, 2010 and *Parawixia dehaani* (Doleschall, 1859) due to the similar abdomen shape and coloration of the three genera (enlarged, oblong with olive brown and light brown markings dorsally on paler area) (Figure 1A and Figure 4A vs. e.g., [38], figures 5A–F; [39], figure 25D). However, the genital morphology of males and females of *Megaraneus* can easily identify this genus from *P. dehaani* and especially from *Backobourkia*. Females of *Megaraneus* and *P. dehaani* differ from *Backobourkia* by having an epigyne with a bulky scape, triangular and not wrinkled, almost completely covering the atrium (Figure 4C; [33], figures 6D,H, 7A, 10D, 12D; [39], figure 3J), while males have pedipalps lacking a bubble-shaped terminal apophysis and the typical flange on the median apophysis of *Backobourkia* (Figure 5C and Figure 6C; [33], figures 6A, 10A, 12A; [40], figure 290g). In females, *Megaraneus* is distinguished from *P. dehaani* by the scape of the epigyne uniformly tapering to its tip, which ventrally overreaches the epigyne border by around half of the scape’s length (vs. scape very wide anteriorly, abruptly tapering to a thin tip that overreaches the epigyne border by less than half of the scape’s length) (Figure 4C,D; [39], figure 3J; [40], figure 290d,e). Finally, males of *Megaraneus* are distinguished from those of *P. dehaani* by the pedipalp with a larger conductor and conductor lobe, and by the presence of a strong triangular protrusion on the base of median apophysis. In contrast, *P. dehaani* males exhibit a smaller conductor and conductor lobe, and median apophysis without protrusions (Figure 5C and Figure 6C; [40], figure 290g).
** **
** **
**Description**. Large orb-weaving spiders, with females (ca. TL 20.2–29.1) between 3.4 and 5.1 times larger than males (ca. TL 5.7–5.9). Preserved specimens with carapace longer than wide, pear-shaped; reddish brown with dusky sides and numerous setae with cup-shaped bases (Figure 4A and Figure 5A). Living specimens sometimes with a black carapace (Figure 1A). Fovea transverse (Figure 4A and Figure 5A). Row of posterior eyes slightly recurved, lateral eyes almost touching on a small tubercle; anterior median eyes slightly protruding from the carapace in both sexes (Figure 4A and Figure 5A). Sternum longer than wide, orange to reddish brown with dark contour and mottled beige (Figure 4B and Figure 5B). Labium wider than long, dark brown with anterior pale edge (Figure 4B and Figure 5B). Maxillae dark brown with pale antero-mesal section (Figure 4B and Figure 5B). Chelicerae dark brown. Leg formula IV > I > II > III for females and I > IV > II > III for males. Abdomen oblong, elongate cylindrical, posteriorly much narrower, with six strong humps (Figure 4A and Figure 5A); female with projected posterior end, rounded in males (Figure 4A and Figure 5A); at least two pairs of conspicuous black sigillae, the central ones larger (Figure 4A and Figure 5A); abdomen otherwise without specialized setae, condyles or other specific structures; females dorsally with brown background exposed anteriorly and followed by a large golden brown area, while males dorsally with olive gray background, medially lighter (Figure 4A and Figure 5A). Venter of females golden brown with two parallel black marks and other sparse black dots; venter of males olive grey to black (Figure 4B and Figure 5B). Female epigyne subquadrate, atrium centrally concave with excavated sclerotized sides (Figure 4C); scape broad, elongated, covering most of the atrium, and reaching over the epigyne base (Figure 4C,D); posterior plate with strong sclerotized sides, slightly tapering posteriorly with central division narrow (Figure 4E); internal genitalia with spherical enlarged spermathecae occupying most of the epigyne area (Figure 6A,B). Male pedipalp patella with a single short macroseta (Figure 5C); tibia with two strong macrosetae, only socket visible dorsally (Figure 5C,D); paracymbium very short with rounded tip (Figure 5D); median apophysis relatively short, medially enlarged with rounded tip, bearing an extremely enlarged and triangular protrusion on its base (Figure 5C and Figure 6C); radix wide and curved (Figure 5C and Figure 6C); stipes inconspicuous (Figure 5C and Figure 6C); terminal apophysis elongated, filiform, straight and transparent (Figure 5C,D and Figure 6C,D); conductor lobe enlarged and triangular, with a sclerotized base bearing scale-like structures (Figure 5C); conductor prominent, spoon-shaped, centrally fleshy and slightly concave, projected retrolaterally as two pointed and sclerotized tips (Figure 5C and Figure 6C); embolus basally broad, tapering apically to an elongated and filiform uncapped tip (Figure 5C and Figure 6C).
** **
** **
**Distribution**. According to the World Spider Catalogue, specimens of *Megaraneus gabonensis* have so far been recorded in Sierra Leone, Cameroon, Gabon, Republic of Congo, Angola, Mozambique, and South Africa [41]. For an up-to-date species occurrence map, see GBIF [42]. Our specimens were collected in iSimangaliso Wetland Park, South Africa.
** **
** **

***Megaraneus gabonensis* (Lucas, 1858)**
Figure 1, Figure 4, Figure 5 and Figure 6.
** **
** **
*Epeira gabonensis* Lucas [43]: 420–422, pl. 12, figure 6.*Epeira angolensis* Brito Capello [44]: 79–80, pl. 2, figure 4.*Epeira chinchoxensis* Karsch [45]: 333–334.*Aranea basilissa* Thorell [46]: 44.*Cyrtophora angolensis* Pocock [47]: 853–854.*Cyrtophora gabonensis* Simon [48]: 286.*Caerostris basilissa* Roewer [49]: 888.*Megaraneus campbelli* Lawrence [18]: 110–115, figures 1a–e and 2a–e,g (first male description).*Megaraneus gabonensis* Grasshoff [50]: 763–764.*Megaraneus gabonensis* Dippenaar-Schoeman et al. [51]: 47–48, p. 47: 1 figure, p. 48: 6 figures.
** **
** **
**Type-material.** Holotype female of *Epeira gabonensis* Lucas, 1858 from Gabon, western Africa coast (MNHN?), not examined.Female syntype of *Epeira angolensis* Brito Capello, 1866 from Rio Cuilo (Kwilo or Quilo), Angola/Democratic Republic of Congo, José de Anchieta coll. (ACL?), not examined.4 syntype females of *Epeira chinchoxensis* Karsch, 1879 from Chinchoxo (Tschinschotscho), Cabinda, Angola, Falkenstein J. coll. (ZMB 2950), not examined.Female syntype of *Aranea basilissa* Thorell, 1899 from Cameroon, Sjöstedt coll. (NHRS?), not examined.Holotype female of *Megaraneus campbelli* Lawrence, 1968, from eastern shore of Lake Sibayi, I. D. J. Jones coll., July 1967 (RU), not examined; male allotype from same locality of holotype, except R. F. Lawrence coll., January1968 (RU), not examined; 2 females and 1 male paratypes, same data as allotype (RU), not examined.
** **
** **
**Material examined.** See Appendix B.
** **
** **
**Diagnosis**. As for genus, *Megaraneus* is monotypic.
** **
** **
**Description**. *Female* (based on WAM T170877): Total length 20.6. Carapace 10.0 long, 8.9 wide; entirely reddish brown with darker sides and a central dark spot at cephalic area (Figure 4A). Eyes: AME 0.56, ALE 0.48, PME 0.53, PLE 0.45. Chelicerae black (Figure 4B). Legs black with femora lighter ventrally near coxae; all legs with sparse yellowish setae (Figure 4A,B). Pedipalp length of segments (femur + patella + tibia + tarsus = total length): 3.4 + 1.7+ 3.0 + 4.5 = 12.6. Leg formula IV > I > II > III; length of segments (femur + patella + tibia + metatarsus + tarsus = total length): I—11.1 + 5.0 + 9.8 + 8.6 + 3.4 = 37.9, II—10.4 + 4.6 + 8.9 + 8.5 + 3.6 = 36.0, III—8.1 + 4.1 + 5.6 + 4.9 + 2.5 = 25.2, IV—12.5 + 5.8 + 9.6 + 8.8 + 3.5 = 40.2. Labium 1.53 long, 2.36 wide, dark brown, almost black, and maxillae dark brown, both anteriorly light brown and pale (Figure 4B). Sternum 5.7 long, 4.7 wide, orange brown with lighter streaks (Figure 4B). Abdomen 10.6 long, 18.6 wide, bearing six humps, three on each side, and a projected and pointed posterior end; dorsally with olive brown background exposed anteriorly, medially and posteriorly strongly covered in golden brown; four pairs of small to median sized brown sigillae (Figure 4A), venter golden brown with two parallel transverse black streaks and sparse black dots, laterally olive brown (Figure 4B). Epigyne and spermathecae descriptions as for genus (Figure 4C–E and Figure 6A,B).
** **
** **
*Male* (based on WAM T170875): Total length 5.9. Carapace 2.8 long, 2.3 wide, chelicerae, sternum 1.2 long, 1.1 wide, labium 0.57 long, 0.42 wide, and maxillae as in females (Figure 5A,B). Eyes: AME 0.23, ALE 0.16, PME 0.20, PLE 0.14. Legs orange brown mottled dark (Figure 5A,B). Leg formula I > II > IV > III; length of segments (femur + patella + tibia + metatarsus + tarsus = total length): I—2.0 + 0.9 + 1.8 + 1.6 + 0.8 = 7.1, II—2.0 + 0.8 + 1.6 + 1.5 + 0.8 = 6.7, III—1.4 + 0.6 + 0.9 + 0.8 + 0.6 = 4.3, IV—1.7 + 0.7 + 1.4 + 1.3 + 0.7 = 5.8. Abdomen 3.1 long, 2.7 wide, with similar humps as females, with a rounded posterior end; dorsally olive gray background, medially lighter, and bearing two pairs of small sigillae (Figure 5A); venter with yellowish brown background covered by a large black patch (Figure 5B). Pedipalp length of segments (femur + patella + tibia + cymbium = total length): 0.6 + 0.3 + 0.2 + 0.8 = 1.9; description as for genus (Figure 5C,D and Figure 6C,D).
** **
** **
**Variation**. Total length females 20.2–29.1 (n = 12). Total length males 5.7–5.9 (n = 2). Live female specimens can vary from entirely olive brown to entirely black; or black with white abdomen (Figure 1B). The white area of the abdomen varies between specimens.
** **
** **
**Natural history.** The orb web of the female (Figure 1C) is suspended among shrubs, on short anchor threads close to vegetation. During the day, the spiders hide in an off-web leaf retreat enforced by silk threads. The web is large, outermost spiral area between 43 and 58 cm wide and between 60 and 75 cm in height, with the upper orb portion of 16 to 24 cm (n = 4 female webs). With the capture area of 0.20 to 0.34 m^2^, hub displacement index of 0.65 to 0.73, and ladder index of 1.29 to 1.40, the webs are typical of large araneid spiders [52,53]. The webs are sparse, typically with 25–29 radii and 25–28 capture spirals along one web axis, always with an open hub. For additional natural history observations, see [18].
** **
** **
**Taxonomic notes.** The taxonomic history of the genus *Megaraneus* and its species *M. gabonensis* is complex [41]. *Megaraneus* was proposed by Lawrence [18] to initially accommodate the species *M. campbelli* Lawrence, 1968, while *Megaraneus gabonensis* was first described by Lucas [43] in the genus *Epeira* Walckenaer, 1805 (today a junior synonym of *Araneus* Clerck, 1757). After its original description, two other species in the genus *Epeira* were described, *E. angolensis* (Brito Capello, 1866) and *E. chinchoxensis* Karsch, 1879, both later transferred to *Cyrtophora* Simon, 1864 and synonymized with *C. citricola* (Forsskål, 1775) [54]. Simon [48] later rejected the synonymy with *C. citricola* and synonymized both species with *E. gabonensis* in *Cyrtophora*, possibly due to the abdominal humps, forming the combination *Cyrtophora gabonensis* (Lucas, 1858).

Finally, in 1984 Grasshoff transferred *Cyrtophora gabonensis* to *Megaraneus* and proposed *M. campbelli*, as well as *Caerostris basilissa* (Thorell, 1899) (initially described as *Aranea basilissa* Thorell, 1899) as junior synonyms of *M. gabonensis*, rendering *Megaraneus* monotypic [50]. The morphology of *M. gabonensis* is more comparable with other ‘backobourkiines’ that display eSSD, such as the genus *Backobourkia* and *P. dehaani* (see diagnosis above), rather than *Cyrtophora* and *Caerostris* Thorell, 1868 by females with a long and wide scape covering the atrium (Figure 4C) and males with a strong median apophysis with an arch over the radix and a very thin and elongated terminal apophysis (Figure 5C,D).


** **
** **


**Distribution**. As for genus, *Megaraneus* is monotypic.

## Figures and Tables

**Figure 1 insects-16-00992-f001:**
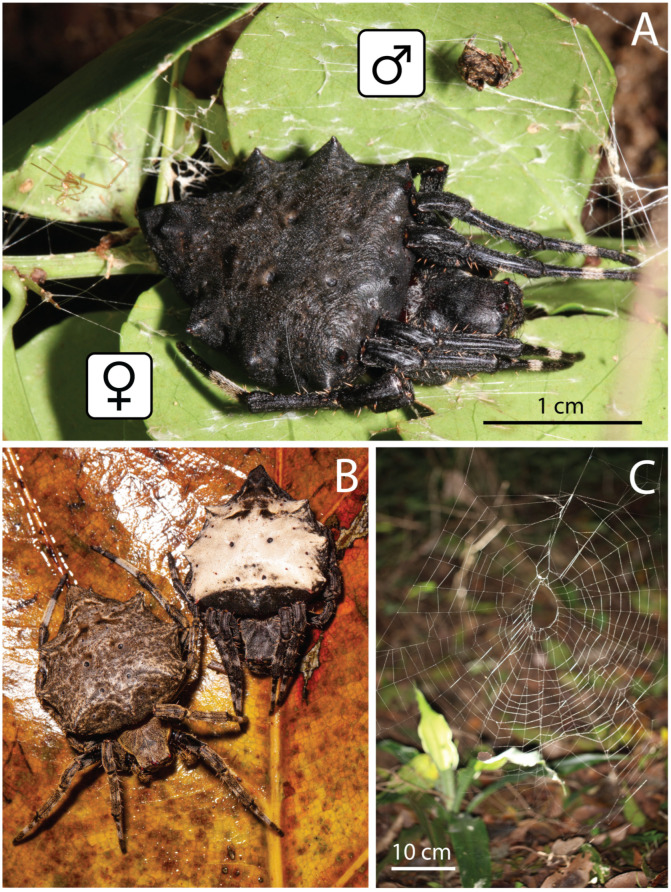
*Megaraneus gabonensis* (Lucas, 1858) from South Africa. (**A**) Male (above) and female (below) resting in a web retreat. (**B**) Color variation in female abdomens. (**C**) Female web. Photos by Matjaž Kuntner.

**Figure 2 insects-16-00992-f002:**
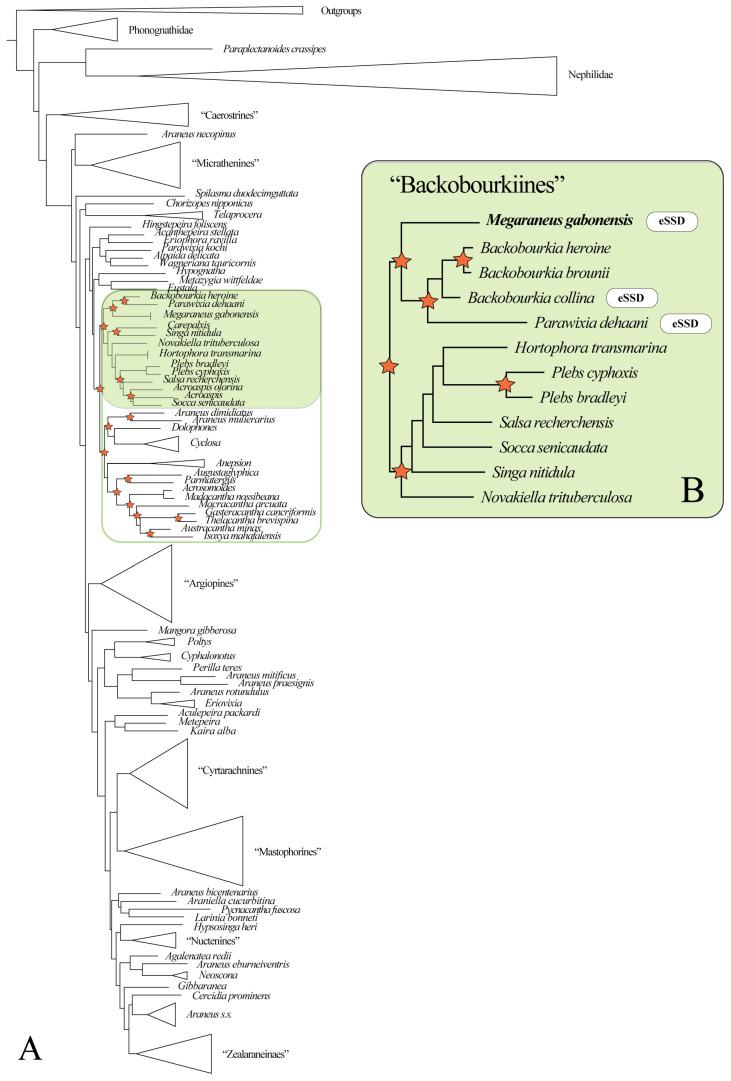
(**A**) Bayesian phylogeny of Araneidae (for full results, see Appendix A). *Megaraneus gabonensis* is placed within the ‘backobourkiines’ (highlighted in green) *sensu* Scharff et al. [20]. Stars indicate nodes with posterior probability > 0.90 within ‘backobourkiines’ and its sister clade (green outline). (**B**) Bayesian reconstruction of a subset of ‘backobourkiines’ (for full results, see Appendix A). Stars indicate nodes with posterior probability > 0.90.

**Figure 3 insects-16-00992-f003:**
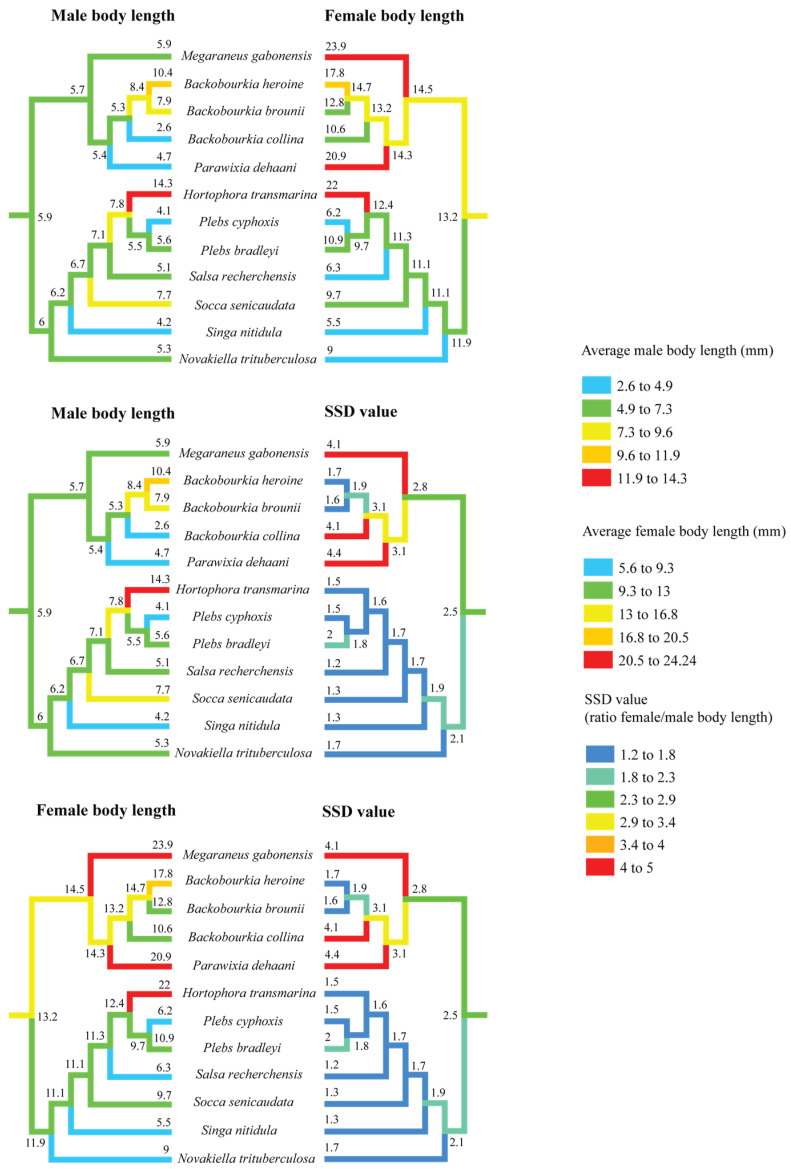
Ancestral state reconstruction of body length and sexual size dimorphism (SSD) in ‘backobourkiines’ using squared-change parsimony. The results demonstrate the complexity of evolutionary change in male and female size that are often decoupled and combined produce SSD.

**Figure 4 insects-16-00992-f004:**
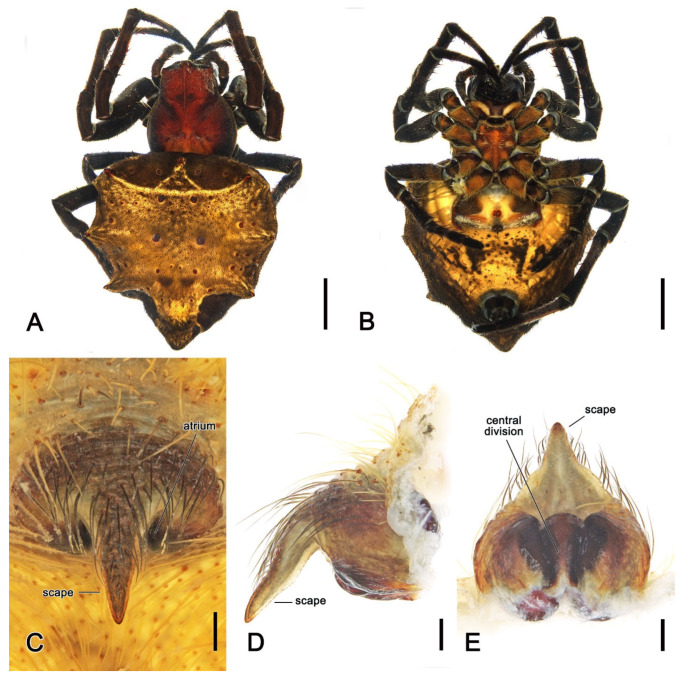
*Megaraneus gabonensis*, female (WAM T170877). (**A**) dorsal habitus; (**B**) ventral habitus; (**C**) epigyne, ventral view; (**D**) epigyne lateral view; (**E**) epigyne posterior view. Scale bars: (**A**,**B**) 2 mm; (**C**–**E**) 0.2 mm.

**Figure 5 insects-16-00992-f005:**
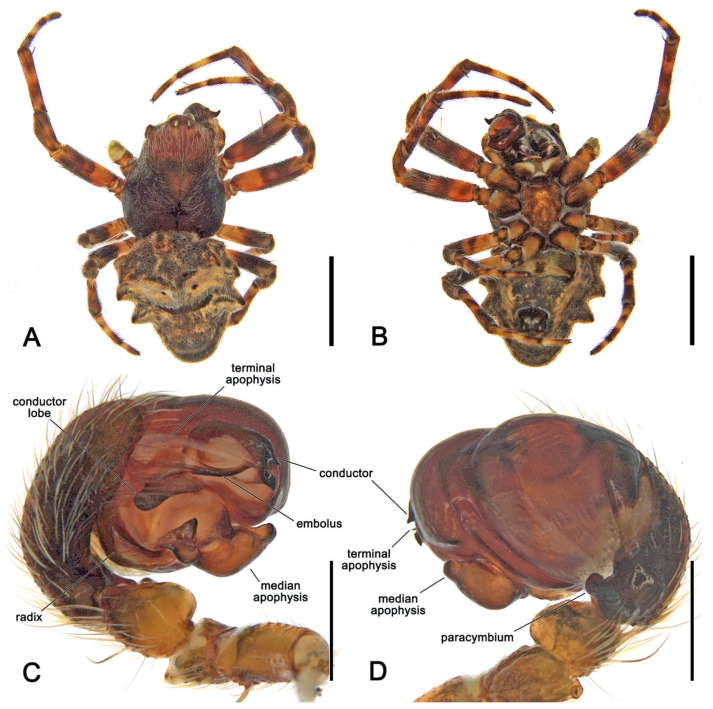
*Megaraneus gabonensis*, male (WAM T170875). (**A**) dorsal habitus; (**B**) ventral habitus; (**C**) left pedipalp, ventral view; (**D**) left pedipalp, dorsal view. Scale bars: (**A**,**B**), 2 mm; (**C**,**D**) 0.2 mm.

**Figure 6 insects-16-00992-f006:**
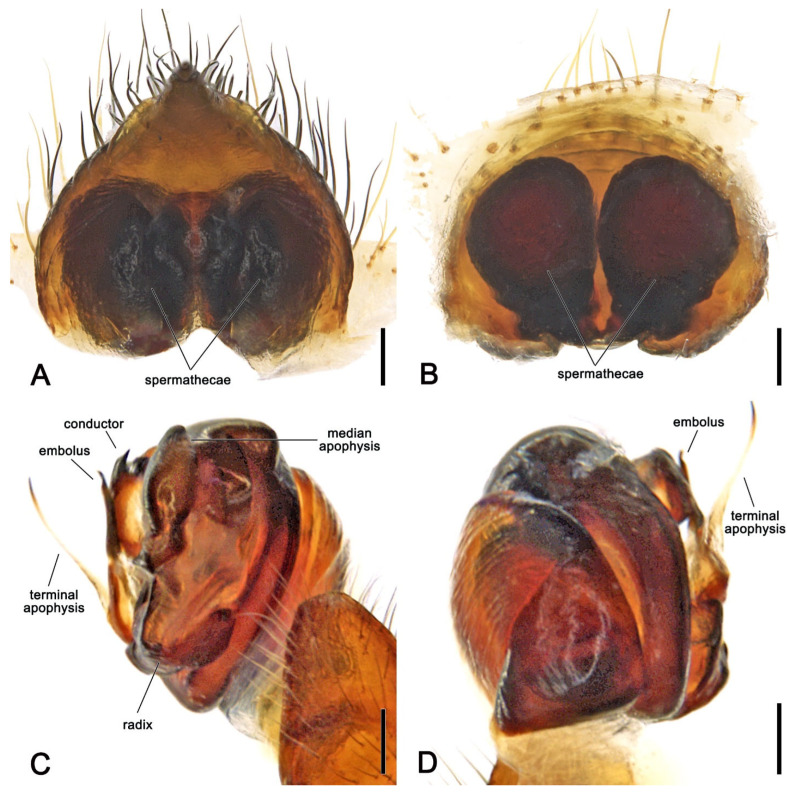
*Megaraneus gabonensis*, female internal genitalia cleared (WAM T170877). (**A**) posterior view; (**B**) anterior view. *Megaraneus gabonensis*, male left pedipalp expanded (WAM T170875). (**C**) baso-ventral view; (**D**) prolateral-dorsal view. Scale bars: (**A**–**D**) 0.2 mm.

## Data Availability

Original sequence data presented in the study is openly available on GenBank. For accession numbers, refer to Appendix B.

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
