# Peer review of "Taxonomy, Phylogeny, and Size Evolution in the Spider Genus *Megaraneus* Lawrence, 1968 (Araneae: Araneidae)"

_insects, 2025, doi:10.3390/insects16100992_

Round 1

Reviewer 1 Report

Comments and Suggestions for Authors

Dear Editor and Authors,

The study present new taxonomic, phylogentic and natural history data regarding  of a poorly known spider. The study is well presented, and well written.

Please see main comments below and detailed comments in the text.

1) The diagnosis needs to be re-written. Please present a comparative diagnosis and refer to figure for compared genera.

2) Please add further morphologic detailed of the female internal genitalia in the description.

Author Response

Reviewer 1

Dear Editor and Authors,

The study present new taxonomic, phylogentic and natural history data regarding  of a poorly known spider. The study is well presented, and well written.

Please see main comments below and detailed comments in the text.

1) The diagnosis needs to be re-written. Please present a comparative diagnosis and refer to figure for compared genera.

Thank you for your comment. We followed the reviewer’s PDF comments and provided additional comparative details of morphological diagnosis and referred to relevant images of related species in other publications. We trust our diagnosis now allows for robust and reliable species identification.

2) Please add further morphologic detailed of the female internal genitalia in the description.

Additional details were provided.

Reviewer 2 Report

Comments and Suggestions for Authors

The part of taxonomy and the phylogenetic analysis are very helpful for researchers and students who is interested in the family Araneidae and surrounging groups. Regarding the change of body size and the gigantism of females, the conclusion seems a bit simplistic. I am sure there is a lot of accumulated research going on, but it would be even more intersting to learn matters like the life cycle or at what point males and females can be (also genetically) distinguished.

Author Response

Reviewer 2

The part of taxonomy and the phylogenetic analysis are very helpful for researchers and students who is interested in the family Araneidae and surrounging groups. Regarding the change of body size and the gigantism of females, the conclusion seems a bit simplistic. I am sure there is a lot of accumulated research going on, but it would be even more intersting to learn matters like the life cycle or at what point males and females can be (also genetically) distinguished.

We thank the reviewer for their positive assessment of our work. While we agree that these are interesting topics for further research, they were out of scope for our brief field survey. All natural history that we were able to observe is documented in the Natural history section of Taxonomy. We now also refer to the original description of the species (Lawrence, 1968) for more natural history observations.

Reviewer 3 Report

Comments and Suggestions for Authors

Only several comments:

Please include references for the software tools, methodologies, and primers used, such as Leica Application Suite X and Geneious.

If possible, please add a distribution map for M. gabonensis.

Author Response

Reviewer 3

Please include references for the software tools, methodologies, and primers used, such as Leica Application Suite X and Geneious.

We thank the reviewer for their comment. Missing software versions were added and we now clearly state that the entire PCR protocol, including choice of primers, followed the procedure from Yu et al. (2022, Invertebrate Systematics) (Lines 135-136).

If possible, please add a distribution map for M. gabonensis.

We appreciate the reviewer’s suggestion. However, the species is currently known from only a few confirmed collection points across the African continent. Given this sparse and highly discontinuous sampling, we believe a distribution map could wrongly suggest a broader or more continuous occurrence than exists in nature. We thus do not provide a map but now instead refer the reader to an up-to-date species occurrence map at GBIF (Lines 360-361).

Reviewer 4 Report

Comments and Suggestions for Authors

This manuscript provides a redescription of the  /Megaraneus gaboensis/, provides evidence of its phylogentic affinities from several molecular molecular markers and explores the evolution of extreme sexual size dimorphism (eSSD) in the context of closely allied "backobourkiines".

In the context of the lack of resolution/stability in the phylogeny  of araneid this study does advance our understanding of this diverse clade, besides adding one more terminal to an already  tangled phylogeny.  I hope the  following general comments be helpful for redesigning a future version of this manuscript.

(1) It is not clear from the introduction/description why M. gaboensis necessitates redescription. Males and females are/there  identity/diagnosis  problems? Note that no type material was examined.

(2) The taxonomic history can be better explained. In the current manuscript it is only tangentially treated in teh cotext of the eSSD? how were males and females associated? contrast/disciuss  the similarities  with similar genera for example te resemblace with "Acanthepeira?".

(3) Critically address the importance and relevance of taxon sampling for resolving the affinities of the focal species within backobourkiines, including relatively closely related Eriophorines.

(4) The authors recognize the difficulties to assess eSSD in consideration of sample sizes, intra-specific variation and taxonomic sampling. Nevertheless, there is no discussion or explanation about how those limitation complexities were  addressed, if at all.

5) Justify the use squared-change parsimony  over other continuous trait ancestral character reconstruction algorithms.  Discuss the effect of choice of outgroup.

Other minor comments:

-  Line 106: The final objective contradicts the argument in line 62-63, about the complexity of the eSSD phenomenon.

  • Figure 3 remove inset male vs female body length (mm). The information is already provided in the previous insets an the figure will benefit from larger scale to improve readability.
  • -Methods and Supplementary table. Indicate the sample size for the averages reported. For example, for Backobourkia brounii the length data shows an interval (8,75 - 16,88) and the average reported is the arithmetic mean of these extremes (12,815)! this is only true if  n = 2. Please clarify.

Author Response

Reviewer 4

This manuscript provides a redescription of the  /Megaraneus gaboensis/, provides evidence of its phylogentic affinities from several molecular molecular markers and explores the evolution of extreme sexual size dimorphism (eSSD) in the context of closely allied "backobourkiines".

In the context of the lack of resolution/stability in the phylogeny  of araneid this study does advance our understanding of this diverse clade, besides adding one more terminal to an already  tangled phylogeny.  I hope the  following general comments be helpful for redesigning a future version of this manuscript.

(1) It is not clear from the introduction/description why M. gaboensis necessitates redescription. Males and females are/there  identity/diagnosis  problems? Note that no type material was examined.

We thank the reviewer for their favorable comments. While we acknowledge previous descriptions of the genus and species, a major taxonomic contribution of our present description is the inclusion of microscopy imaging, showing detailed morphological features for species diagnosis. The last valid taxonomic description dates from 1968, which is by today’s standards outdated. We therefore provide a modern redescription of the species.

(2) The taxonomic history can be better explained. In the current manuscript it is only tangentially treated in teh cotext of the eSSD? how were males and females associated? contrast/disciuss  the similarities  with similar genera for example te resemblace with "Acanthepeira?".

The taxonomic history is listed in the Taxonomy section; however, we are unsure of what the comment refers to in its relation to eSSD. The males and females were identified as representatives of the same species by their morphology (in line with previous descriptions), genetics (via COI sequences) and behavior (occurring on the same web). According to the phylogeny reconstructed in our study and in Scharff et al. (2020), ‘backobourkiines’ (and with it Megaraneus) are not part of the same clade as Acanthepeira and thus no direct comparison was made. However, we did compare Megaraneus to other members of ‘backobourkiines’, namely Parawixia dehaani and Backobourkia.

(3) Critically address the importance and relevance of taxon sampling for resolving the affinities of the focal species within backobourkiines, including relatively closely related Eriophorines.

Please see the comment above.

(4) The authors recognize the difficulties to assess eSSD in consideration of sample sizes, intra-specific variation and taxonomic sampling. Nevertheless, there is no discussion or explanation about how those limitation complexities were  addressed, if at all.

Indeed, limited sample sizes are a common problem of comparative analyses of SSD, to which our study was not immune. Our data on Megaraneus includes all available specimens, and we have combed the available literature for size data on the related genera.

5) Justify the use squared-change parsimony  over other continuous trait ancestral character reconstruction algorithms.  Discuss the effect of choice of outgroup.

We used squared-change parsimony because it requires no assumptions about the underlying statistical distribution of trait evolution. Our focus was on reconstructing general trends in trait evolution rather than making precise probabilistic estimates and squared-change parsimony serves this purpose effectively. We now provide a short justification for this in the text (Lines 186-188).

Other minor comments:

-  Line 106: The final objective contradicts the argument in line 62-63, about the complexity of the eSSD phenomenon.

We appreciate the reviewer’s comment. We rephrased both parts to make our argument clearer and avoid contradictions.

- Figure 3 remove inset male vs female body length (mm). The information is already provided in the previous insets an the figure will benefit from larger scale to improve readability.

We appreciate the comment on information redundancy; however, sex-dependent trajectories of size evolution are a major topic addressed in the paper and we believe a mirrored image of male and female size evolution is nevertheless visually informative. We do agree that the readability of the figure could be improved and have changed the configuration and font sizes in the figure.

- Methods and Supplementary table. Indicate the sample size for the averages reported. For example, for Backobourkia brounii the length data shows an interval (8,75 - 16,88) and the average reported is the arithmetic mean of these extremes (12,815)! this is only true if  n = 2. Please clarify.

We thank the reviewer for noticing this oversight. In response, we have thoroughly revisited all the papers used for size calculations. We now include the number of specimens used to calculate averages reported in the literature whenever possible. In cases where only size intervals were available, we had to rely on the arithmetic mean only. Using these updated averages, we reanalyzed the ancestral character reconstruction. Accordingly, we have revised the supplementary table and updated Figure 3 to reflect these changes.

Round 2

Reviewer 4 Report

Comments and Suggestions for Authors

The major concerns raised in the previous manuscript remain unchanged in this iteration. 

(1) There is little justification for the redescription of this species. The argument of the outdated description is insufficient. 

(2) In the context of a poorly resolved, undersampled and  unstable phylogeny, I find it unreasonable to completely ignore the affinities previously suggested for this taxon. Even if the ‘backobourkiines’ hypothesis holds, the resemblance to other (more or less) closely related araneid is worth discussing.

(3) There is no critical commentary on the effects of sampling (intra- and inter- specific )on the ancestral character reconstruction  and the conclusions derived from those estimates.  I understand all available data was used but that alone does not warranty the estimates to be less biased.

Author Response

The major concerns raised in the previous manuscript remain unchanged in this iteration.

(1) There is little justification for the redescription of this species. The argument of the outdated description is insufficient.

We thank the reviewer for their comment. As we now also state in the manuscript (see rewritten Discussion), prior description of the genus and species is 57 years old and is, by today’s standards, outdated. We therefore provide a redescription that adheres to modern standards in araneology. The major taxonomic contribution of our present description is the inclusion of microscopic imaging that shows detailed morphological features used in genus and species diagnoses. Importantly, the differential diagnosis of the genus that we provide emphasizes the phylogenetic proximity of Megaraneus as revealed by our phylogenetic analyses, the first to precisely place the genus.

(2) In the context of a poorly resolved, undersampled and  unstable phylogeny, I find it unreasonable to completely ignore the affinities previously suggested for this taxon. Even if the ‘backobourkiines’ hypothesis holds, the resemblance to other (more or less) closely related araneid is worth discussing.

Despite previous taxonomic placements of the species, molecular data now strongly suggest that Megaraneus does in fact place within ‘backobourkiines’ and is not closely related to the genera previously believed to contain this species (e.g. Eriophora, Cyrtophora). Our conclusions are now further corroborated by a maximum likelihood analysis on a subset of Araneidae taxa. Nevertheless, we added a section Taxonomic notes to the manuscript, addressing dissimilarities with genera, previously believed to be closely related.

(3) There is no critical commentary on the effects of sampling (intra- and inter- specific )on the ancestral character reconstruction  and the conclusions derived from those estimates.  I understand all available data was used but that alone does not warranty the estimates to be less biased.

We added a cautionary commentary to the Discussion section regarding this issue.